# From the Order of Zong Fa (宗法) to the Order of Ren Lun (人倫)—Confucianism and the Transformation of the Paradigm of Early Chinese Communities

**Yun Chen** [1,2]

1   Institute of Modern Chinese Thought and Culture, East China Normal University, Shanghai 200241, China; ecnuchenyun@163.com
2   Ma Yifu Academy, Zhejiang University, Hangzhou 310030, China

**Abstract:** The form of community established by early Confucianism, represented by Confucius and Mencius, can be called the Ren Lun (人倫) community. This community order contains two interdependent dimensions: at the ethical level, it is primarily dominated by the parent-child relationship, with filial piety as its core dimension; at the moral level, its essence lies in the consciousness of human nature centered around Ren (仁), Yi (義), Li (禮) and Zhi (智) as its core, which goes beyond a mere universal human nature. This differs from the order of the Zong Fa (宗法) community in the Western Zhou Dynasty, whose axis is the way of brotherhood, with a vertical lineage connecting ancestors and descendants, in order to achieve unity and cohesion among the horizontal brother tribes. In the Zong Fa (宗法) community, morality and ethics are undifferentiated, and there is no distinction between individual and collective virtues, as well as ruling virtues and edifying virtues. The spiritual principle of the Zong Fa (宗法) community is Qin Qin Zun Zun (親親尊尊), which is both continuous and different from Ren Yi (仁義), revered by early Confucianism. Ren Yi (仁義) is extracted from Qin Qin Zun Zun (親親尊尊), but as a value principle, it possesses a higher universality. Qin Qin Zun Zun (親親尊尊) is a systemic principle closely tied to Zhou Li (周禮), while Ren Yi transcends the system as independent moral principles.

**Keywords:** Zong Fa (宗法); ethics; Qin Qin (親親) and Zun Zun (尊尊); Ren Yin (仁義)

## 1. Introduction

Early Confucianism, established by figures like Confucius and Mencius, laid the foundation for the dominant order in traditional Chinese society for over two thousand years, from the Qin Dynasty to the Qing Dynasty. When we say that traditional Chinese society is based on ethics, it refers to the order of Ren Lun (人倫). As a kind of prototype of orders, the order of Ren Lun (人倫) is distinct from the Greek order of the polis, the Hebrew order of revelation and the modern order of "social" centered around individual aggregation. The fundamental relationships in this order of Ren Lun (人倫) are the Five Relationships (五倫): "the relations between parents and children there is affection; between ruler and minister, rightness; between husband and wife, separate functions; between older and younger, proper order; and between friends, faithfulness" (父子有親，君臣有義，夫婦有別，長幼有序，朋友有信)" (*Mencius*, "Teng Wang Gong"I,《孟子·滕文公上》), which summarizes the essence of this order. However, tracing the origin of this ethical paradigm requires examining the social and historical conditions in which it emerged and the soil on which the order was established. This leads us back to the Zhous order of Zong Fa (宗法). Whereas the transition from the order of Zong Fa (宗法) to the order of Ren Lun (人倫) involves the differences between the way of Duke Zhou and the way of Confucius, the most significant aspect is the transformation from the Three Dynasties' paradigm of order with Zhou as the main body to the new paradigm initiated by Confucius.

**2. The Order of Zong Fa (宗法) Community: The Way of Elder Brothers Leading Younger Brothers**

The evolution from lineage to clan constituted a key transition phase in the transformation of society from the Xia and Shang dynasties to the Western Zhou Dynasty (Chao 1996, pp. 233, 235, 255, 277; Y. Chen 2014, pp. 129–40). With the development of lineage society, the population continued to increase, leading to the differentiation of branches within lineage organizations. In the context of a vast number of states coexisting where each state represented a distinct lineage, the unity and integration among the branch lineages faced challenges. The intensification of lineage differentiation further exacerbated this issue. The series of institutions, including the order of Zong Fa (宗法) in the Zhou Dynasty, emerged in response to the natural differentiation within lineage organizations, which resulted in the weakening of overall collective strength. In the political structure of the state of Zhou (known as the "Inner State" or "Zhong Guo中國") and the "myriad states" (known as the "Outer States" or "myriad states" or "vassal states"), the ruling Zhou clan faced not only continuous natural differentiation within the clan organization but also the necessity of recognizing the existing situation of "one state" among "myriad states" by adopting a "feudal" system, which itself further reinforced the differentiation of the Zhou clan.

In order to counteract the tendency of differentiation and political centrifugal forces caused by differentiation, the people of the Zhou Dynasty employed various systems such as Zong Fa (宗法) and the act of bestowing surnames. These measures aimed to bring fragmented clan organizations back under the control of the ruling group, achieving the greatest possible unity within the ruling elite. This emphasis on unity was repeatedly emphasized by the people of the Zhou Dynasty as they learned from the lessons of the downfall of the Shang Dynasty. Chen Jie astutely pointed out that when the Zhou people, during the natural differentiation of kinship groups, used systems such as bestowing surnames to achieve cohesion within the clan, "this is precisely the key difference between the differentiation of lineages within a clan society and a kinship society." Once the institutions of orders, such as the order of Zong Fa (宗法), were established, the social structure underwent a transformation. The natural differentiation and branching of lineage organizations based on the principle of equality were reshaped into hierarchical divisions within clans (J. Chen 2007, pp. 293–94). Both in the lineage society and clan society, various sizes of kinship groups existed based on blood relations, which determined the closeness or distance among individuals.

However, the distinguishing feature of clan society lies in its structure. Through the continuous ritual activities and the presence of a shared ancestor as a symbol, the differentiated lineages within the clan are integrated as distinct "branches" on the same "clan tree." This integration is achieved by the transmission of ancestral symbols through the legitimate eldest son of the clan (嫡長子), who becomes the bearer of the ancestral symbol.

Thus, it forms the backbone of the practical order of Zong Fa (宗法), where "the family members" (宗人) govern "all clan members" (族人). Through the integration of ancestor worship and the legitimate eldest son, the emphasis is placed on the rightful inheritance of the ancestral lands, people and ancestral temple by the legitimate eldest son. This highlights the esteemed position of the legitimate eldest son and establishes a hierarchical relationship between the legitimate eldest son and other members of the clan, emphasizing their respective positions of honor and subordination.

As a result, the legitimate eldest son becomes the subject of liability for "bringing together the clan" (uniting and consolidating the clan 收族). Therefore, in the order of Zong Fa (宗法), all members within a clan are of the same lineage and constitute integral parts of the same clan community. Under the leadership of the legitimate eldest son, individuals can take care of each other's good fortune and misfortune, exchange surplus goods to obtain what they lack, maintain order and hierarchy between the young and the old, differentiate between close and distant relationships without complete separation and have different social statuses without interfering in each other's affairs. The act of "bringing to-

gether the clan" (收族) creates a sense of kinship within the same clan, making individuals from the same clan appear as if they are from the same family, and individuals from the same family appear as if they are one entity.[1]

Therefore, in the clan society established during the Zhou Dynasty, the relationship between numerous sub-clans derived from the same mother clan ceased to be equal. Instead, it became a system where a revered individual (the legitimate eldest son) (宗子) and the people who revered him (clan members) (族人) were established through the system of primogeniture. The establishment of the system of major and minor lineages (大小宗制度) led to the overarching authority of the major lineage over numerous minor lineages. Within the minor lineages, distinctions were made between lineages that traced their ancestry back to the great-grandfather, the great-great-grandfather, the grandfather, and the great-uncle, each having their own lineage leader to oversee them.

This formed a hierarchical and differentiated structure of leadership, resembling a branching tree, with varying levels of authority. This structure represents a form of unity among kinship groups that is integrated from smaller units to larger ones, building upon each other in a bottom-up manner. It overall enhances the Zhou people's ability to consolidate and unite through institutionalized systems like Zong Fa (宗法), thus mitigating or even resisting the weakening of state power caused by the natural proliferation of lineage branches. Indeed, facing the trend of lineage society's proliferation and the diminishing cohesive force of blood ties as the society grew larger, the Zhou people had to transform the original lineage-based organization into the institutionalized system of the kinship community of Zong Fa (Guan 2010, p. 28).

The Zong Fa (宗法) communal system utilizes blood relationships to integrate clan members into the order of the Zong Fa (宗法) community based on the principle of Qin Qin (親親). However, Qin Qin (親親) is not the sole principle of the order of Zong Fa (宗法). Relatively speaking, the principle of Zun Zun (尊尊) is the core principle of the Zong Fa (宗法) community. In terms of the spontaneous evolution from lineage to clan, the continuous division of lineages provides the foundation and possibility for the establishment of minor lineages.

However, from the perspective of the legislator's creation, it is only through the establishment of major lineages that the branching of lineages can separate from the structure of lineage society and become minor lineages in the Zong Fa (宗法) community. This is because the leadership of the major lineage over the pluralistic minor lineages and clan members is no longer solely based on kinship (親親之仁), but rather on reverence for ancestors. With the inclusion of the concept of righteousness (Yi 義), the process of receiving members into the major lineage extends beyond the scope of kinship and blood relations. Only the major lineage transcends pure kinship and blood relations while still encompassing them. In other words, it combines the principles of kinship (親親之義) and political reverence (尊尊之義). It is through this combination that Zong Fa (宗法) became the shaping principle of a political-ethical community.

The principle of Qin Qin (親親) has limitations and boundaries when it comes to achieving clan cohesion. In the *Book of Rites*, "Sang Fu Xiao Ji" (《禮記•喪服小記》), it is recorded as follows: "Among relatives with blood ties, the closest are the father above and the son below. From these three generations of relatives, it expands to five generations, including the grandfather above and the grandson below. It further expands to nine generations, including the great-grandfather above and the great-grandson below. The degree of mourning attire is arranged based on these degrees of kinship, with each generation reducing mourning attire from the father upwards and from the son downwards. As for non-direct lineage relatives, the more distant the blood relationship, the greater the reduction in mourning attire until there is no familial bond."[2] The sequence of blood relations, starting from oneself, includes the father above and the children below, collectively forming three generations. It expands further by extending upward to the grandfather and downward to the grandchildren, thus expanding from three generations to five generations. It further expands by extending to the great-grandfather and great-great-grandfather through the

lineage of the grandfather, and to the great-grandchildren and great-great-grandchildren, thus expanding from five generations to nine generations. This represents the hierarchical pattern of kinship based on Qin Qin (親親). Starting from oneself, the kinship sequence expands from three (father-self-child) to five (grandfather-father-self-child-grandchild) to nine (great-great-grandfather-great-grandfather-grandfather-father-self-child-grandchild-great-grandchild-great-great-grandchild). As the kinship becomes more distant, the mourning attire gradually becomes lighter until it is no longer worn.

Indeed, it can be observed that from an individual's perspective, the expansion of kinship extends from oneself to father and children and further expands to include grandfather and grandchildren. In this process, the principle of Qin Qin (親親) is followed, gradually extending from close relationships to more distant ones. From the perspective of the Zong Fa (宗法) system, individuals within a clan should revere the legitimate eldest son of the major lineage (大宗宗子) who shares a common progenitor, as well as the legitimate eldest sons who share the same great-great-grandfather, great-grandfather, grandfather and father—collectively referred to as the legitimate eldest sons of the four generations (四類小宗宗子). The principle of Qin Qin (親親) applies to minor lineages, but when considering the distance of kinship, after six generations, individuals become like strangers to each other.

Therefore, the saying goes, "After five generations, ancestors will be excluded from the scope of worship (五世則遷)." However, within the major lineage, the principle is based on showing respect to the noble ones (尊尊). With the addition of the principle of Zun Zun (尊尊), treating others with the principle of Zun Zun (尊尊) and collectively honoring sons other than the legitimate eldest son as the legitimate eldest son itself, the major lineage does not dissipate even when the close blood ties come to an end.

As a result, it can endure for many generations without undergoing changes (百世而不遷). At the point where the kinship ties within the minor lineage reach their limit, the major lineage can still fulfill its function of uniting the clan (收族). Therefore, as Wang Fuzhi王夫之 mentioned, "Ancestors who were once included in the ancestral worship of a minor lineage, if they exceed four generations, will be excluded from the scope of worship within the minor lineage. Their worship will be limited to the ancestral worship of the major lineage (過是則遷，唯統於大宗耳)" (F. Wang 1996, p. 833).[3]

In the order of the Zong Fa (宗法) community, the primary subject is actually the sibling relationship. In fact, the order of Zong Fa (宗法) lies exact in the relationship among brothers, which entails the hierarchical leadership of younger brothers and clan members by the eldest son of the main branch in both major and minor lineages. Cheng Yaotian astutely pointed out: "The way of the clan is the way of brothers. In the households of nobles and scholars, it is the way of elder brothers to govern younger brothers, and the way of younger brothers to serve and respect elder brothers" (Cheng 2008, p. 137). The term "Zong" (宗) itself means "master" or "chief." Therefore, "Ji Bie Wei Zong" (繼別為宗) refers to the situation where the person who inherits his father's position and is revered as an ancestor by his descendants becomes the common master of the younger brothers. This person, as an ancestor revered by his own descendants, is regarded as a father figure, just like his brothers who are also revered as fathers by their own sons. Among their children, each has an eldest son who inherits their respective father's position and is revered by their younger brothers. This is what is called a minor lineage. As for the many descendants who inherit the position, they are revered as ancestors by the descendants who inherit the position of the common master. All minor lineages lead their younger brothers and revere them, and this continues from generation to generation (Cheng 2008, p. 137).

This implies that, while the typical understanding of Zong Fa (宗法) focuses on the vertical relationships between successive generations, in practice, it primarily deals with the relationships among brothers of the same generation but different lineages. The degree of closeness or distance between brothers is determined by the number of lineages they are removed from a common ancestor, such as a shared father, grandfather, great-grandfather and so on. This results in varying degrees of kinship, such as full brothers,

cousins (from paternal or maternal side), second cousins, third cousins and so forth, indicating different levels of closeness or distance between them. The more distant the blood relationship between brothers, the more they need to rely on ancestral worship and other means to strengthen their bond through honoring increasingly distant ancestors (from father to grandfather, then from grandfather to great-grandfather and so on, all the way to the founding ancestor). It is the duty of the legitimate eldest sons of both major and minor lineages to carry out these tasks of uniting and maintaining the kinship among brothers of different lineages. They unite the present brothers in the name of their past ancestors, forming different Zong Fa (宗法) communities based on the degrees of blood relationship, whether close or distant.

In this regard, the core of the way of Zong Fa (宗法) lies in the horizontal relationships rather than the vertical ones. It is primarily based on the bonds between brothers rather than on the relationships between fathers and sons. The relationship between major and minor lineages is not a generational one of the same lineage but rather a relationship among brothers based on the distinction between "Di" (嫡 direct line) and "Shu" (庶 collateral line). More accurately, the vertical father–son relationship serves as the background for the horizontal relationship among brothers. It is through the vertical relationship that the horizontal relationships are structured and organized. The focus is on the horizontal relationships rather than solely on the vertical ones (Y. Chen 2019a, pp. 113–205).

The phrase "Filial piety, which emphasizes filial devotion to parents, can be extended to foster brotherly friendship and love" ("孝乎唯孝，友於兄弟")[4] in *Book of History*, "Jun Chen" (《尚書·君陳篇》), reflects the essence of the spirit of the Zong Fa (宗法) society. "Xiao" ("孝") entails treating one's ancestors with respect and care, while "You" ("友") involves treating one's brothers well. In essence, the institution of Zong Fa (宗法) combines the concepts of "Xiao and "You," integrating them through religious ancestral rites that express reverence for ancestors. It further translates the principles of fraternal love into practical rituals and ceremonial practices within the social system of rites and music.

The inscriptions on bronze artifacts record the ideals of "filial piety towards parents and brotherly love towards siblings, only in this manner" ("孝友惟型" ("Li Yi", *Jun*, Volume 2 the second part 《曆彝》,載《攗》卷二之二)), "only when a ruler can both be filial towards parents and befriend his brothers" ("惟辟孝友" (*Shi Qiang Pan*《史牆盤》)) and so on. These writings specifically emphasize the relationship among brothers. In the surviving literature from the Zhou Dynasty, there are also many records that mention the relationship between brothers. For example, "Now this king Ji, in his heart was full of brotherly duty" ("維此王季，因心則友，則友其兄") in *Book of Songs*, "Huang Yi" (《詩經·皇矣》), "Such great criminals are greatly abhorred, and how much more (detestable) are the unfilial and unbrotherly" ("元惡大憝，矧惟不孝不友") in *Book of History*, "Kang Gao" (《尚書·康誥篇》), and so on. During the Zhou Dynasty, it was common for people to use the character "You" ("友") in personal names. Examples include Taishi You (太史友), Neishi You (內史友), Duke Zheng Huan You (鄭桓公友), Duo You (多友) and many others. The word "You" ("友") is indeed associated with the bond of "brotherhood." In the context of the Western Zhou Dynasty, "You" ("友") had the meaning of "brother," referring to kinship brothers outside of one's immediate family. Sometimes, even one's own biological brothers could be included in the designation of "friends" ("朋友") (Zhu 2004, pp. 292–97). The Zuo Commentary (《左传》) recorded the events of the fifteenth year of Duke Wen's reign: Shi Yi had the following statement, saying, 'Brothers should strive individually to achieve perfection. They should assist each other in times of need, celebrate joyous occasions together, offer condolences in times of disaster, pay respectful homage during ceremonies, and express sorrow during funerals. Although their emotions may differ, they should not sever the bonds of friendship and love between them. This is the moral duty towards one's relatives.'. These indicate that the primary relational subject of the order of Zong Fa (宗法) community is primarily the horizontal brotherly relationship.

The system of minor lineages, which emerged during the later period of the Shang Dynasty, already assumed the role of connecting vertical generational relationships, such as "children inheriting their fathers". However, this type of connection was limited. In the Zhou Dynasty, the system of major lineages expanded the scope by tracing back to more distant ancestors, thus incorporating not only brothers, second-order brothers, third-order brothers and their respective small families whose blood ties had already weakened into the same community of Zong Fa (宗法).

This kind of implement elder-brother governance over younger brothers is achieved through two main factors. On one hand, it is accomplished through respecting and honoring ancestors, and on the other hand, through the respect for the legitimate eldest son of the major lineage (大宗宗子). The more ancient the revered ancestors are, the larger the unified family group becomes. The legitimate eldest son of the major lineage serves as the core of this extended family. So, *Book of Rites*, "Sang Fu Xiao Ji" (《禮記●喪服小記》), records: "because they honored the ancestor, they reverenced the Honored Head; their reverencing the Honored Head was the way in which they expressed the honour which they paid to the ancestor and his immediate successor". The purpose of reverencing the Honored Head and honoring the ancestor is not for the sake of the deceased ancestors themselves but rather to foster unity among the living brothers within the major lineage who have grown increasingly distant from one another. *Book of Rites*, "Da Zhuan" (《禮記●大傳》), records: "They honoured the Ancestor, and therefore they reverenced the Head. The reverence showed the significance of that honour". Further, "Where the starting-point was in affection, it began with the father, and ascended by steps to the ancestor. In the consideration of what is right, it began with the ancestor and descended in a natural order to the deceased father. Thus, the course of humanity (in the matter of mourning) was all comprehended in the love for kin. From the affection for parents came the honoring of ancestors; from the honoring of the ancestor came the respect and attention shown to the heads of the family branches. Through this respect and attention to those heads, all the members of the kin were kept together. This unity led to the dignity of the ancestral temple. From that dignity arose the importance attached to the altars of the land and grain. This importance resulted in the love of all the people with their various surnames. From that love came the proper administration of punishments and penalties, leading to a sense of repose among the people. Through that restfulness all resources for expenditure became, sufficient. Through the sufficiency of these, what all desired was realized. The realization led to all courteous usages and good customs; and from these, in fine, came all happiness and enjoyment".

If we consider "Da Zhuan" (《大傳》) as an explanation of the "Sang Fu" (《喪服》), what is the elucidation of the significance of the "Sang Fu" (《喪服》) to the order of Zong Fa (宗法) community? In the chapter "Qi Shuai Qi" of "Sang Fu Zhuan" (《喪服傳》 "齊衰期"), it states, "The major lineage is the supreme and legitimate authority of the clan. Animals recognize their mothers but not their fathers. People in rural areas may ask, 'What significance do parents hold?' However, those in urban areas understand the importance of honoring their fathers. The officials and scholars recognize the importance of honoring their grandfathers. Princes honor their great ancestors, and the emperor honors the origins of his ultimate progenitor. The noble and esteemed individuals extend their reverence through the ancestral lineage to the distant past, while the humble individuals show reverence along the ancestral lineage to the recent past. The major lineage represents the highest and legitimate authority. It serves as the cohesive force for the clan and should not be severed".

In the chapter "Qi Shuai San Yue" of "Sang Fu Zhuan" (《喪服傳》 "齊衰三月"), it states, "Respecting ancestors is the reason for honoring the legitimate eldest son of the major lineage. Honoring the legitimate eldest son of the major lineage is the guideline for respecting ancestors." It is evident that the veneration of ancestors and the respect for the progenitor of the major lineage ultimately revolve around the act of uniting the clan, which is centered around the legitimate eldest son of the major lineage. This purpose de-

termines that the essence of the order of Zong Fa (宗法) community lies in utilizing vertical generational relationships to handle horizontal brotherly relationships. Consequently, the primacy of brotherhood within the order of Zong Fa (宗法) community becomes essential.

The transformation of the order of Zong Fa (宗法) reforms the clan community, reversing the differentiation of the clan into the unity of Zong Fa (宗法) community. However, whether in a clan society or a Zong Fa (宗法) society, individuals are subordinate to the community and do not exist separately from their identity as members of the community. Accordingly, in the Zong Fa (宗法) society, ethics and morality are not differentiated. It simply involves incorporating individuals into the community, and there is no individual consciousness that seeks to separate from the community. Therefore, in the Zong Fa (宗法) society, the consensus achieved through ritual and ceremonial activities permeates the collective unconsciousness of its members. Within it, there is no distinction between becoming an individual and becoming a member of the community. In fact, becoming a member of the Zong Fa (宗法) community is the way of being human, and determining the meaning of individual existence is finding one's place within the "tree of Zong Fa" ("宗法樹").[5]

### 3. The Father–Son Relationship and Filial Piety in the Order of Ren Lun (人倫) Relationships' Community as a Central and Fundamental Axis

The background of the establishment of the Ren Lun (人伦) community is the disintegration of the Zong Fa community. As the creation of the Western Zhou Dynasty, the Zong Fa was closely associated with the feudal system. The feudal system of the Zhou was divided into two levels: first, the king of the Zhou divided the children of the royal family into vassal states, the son of heaven gave them a surname 姓, and the recipients of the title became vassals by acquiring a surname and establishing a state; and second, the vassals were given shi (氏) (a branch of the family name), and the recipients were given land to establish a Aristocrat's House and became ministers. The condition for the possibility of feudalism was that the Zhou people continued to expand, so there was surplus land and population, which could be distributed to the sons and younger brothers of the king; on the other hand, the population to be rewarded was based on the clan. The grantee received a rewarding surname or shi (氏) from a superior ruler and thus became a legitimate eldest son (宗子) in a patriarchal Zong Fa (宗法) community. In other words, the eldest son inheritance system in the feudal system is actually the application of the patriarchal Zong Fa system, so the patriarchal system is also called the Zong Fa feudal system. Once the Zhou had no surplus land and population to distribute, and once the basic unit of society had diverged from the patriarchal clan to the small main family, then the patriarchal community and its ethical form of life became untenable.

In fact, during the Spring and Autumn and Warring States periods in which Confucius and Mencius lived, the patriarchal clan system had declined, and the backbone family, with five or seven members as the main body, constituted the basic unit of society. The monogamous family, formed by ordinary men and women, came to the forefront of history, and with the establishment of the system of the "bianhuqimin" (编户齐民), this kind of small family constitutes the foundation of society. Whereas the Zong Fa system demonstrated the political connection between the old brother's state and the young brother's state, the human relationships within the main family lost their political function of unifying the different states and became an ethical field for social ordering. For example, filial piety was no longer a way for people in different families within the same clan in the patriarchal system to remember their common ancestor, but a way for small families to respect for patrilineal parents of the family (L. Wang 2007, pp. 175–213).

During this period, there was a simultaneous process of territorial expansion by states. In other words, the expansion of nations and the shrinking of families were two different aspects of the new order. When contemplating the social and ethical order, one cannot ignore this structural political and social transformation. The concept of the Ren Lun (人倫)

community, as conceived by Confucius, Mencius and other Confucian scholars, can be understood within this context to some extent.

When the order of Zong Fa (宗法), as embodied by the major-minor lineage system (大小宗制度), is replaced by the nuclear family as the central unit, it signifies a shift in the ethical axis. The core of the Ren Lun (人倫) community based on the nuclear family is the father–son relationship, which is different from the Zong Fa (宗法) community that centers around the brotherhood. Therefore, with the emergence of Confucianism and the teachings of Confucius and Mencius, a change can be observed where the father–son relationship replaces the brotherhood as the core of order construction. In the Zong Fa (宗法) community, while the father–son relationship is important, it primarily serves as the background for the brotherhood. Through tracing the common ancestry of the brothers to a shared paternal ancestor, a greater sense of unity can be established among brothers within the same era. This is the primary concern of the system of the Zong Fa (宗法) community. The nuclear family, on the other hand, does not function as an independent entity but rather as a component of the clan system. The fundamental unit of social order is the clan, not the nuclear family. During the Spring and Autumn and Warring States periods, with the rise of the nuclear family as an independent entity, the fundamental unit of social order shifted from the clan to the family. Adapting to this structural change, the ethical framework centered around the father–son relationship replaced the Zong Fa (宗法 kinship-based) ethics centered around brotherhood. As a result, the father–son relationship transitioned from the background to the foreground, becoming the paramount relationship within human society.

With the foundation for maintaining political and social stability being established upon the family, and with the father–son relationship becoming a central and fundamental axis in ethics, filial piety naturally became the core of the human ethical order. Paternal kindness and filial piety originally constitute a bidirectional and interactive ethical requirement. In this order of Ren Lun (人倫) community, each individual can find their own ethical obligations. For instance, in the father–son relationship, kindness is the father's duty, while filial piety is the duty of the son. The order of Ren Lun (人倫) demands that everyone fulfills their respective obligations, which is an undeniable responsibility. *Book of Rites*, "Li Yun" (《禮記●禮運》), records: "What are 'the things which men consider right?' Kindness on the part of the father, and filial duty on that of the son; gentleness on the part of the elder brother, and obedience on that of the younger; righteousness on the part of the husband, and submission on that of the wife; kindness on the part of elders, and deference on that of juniors; with benevolence on the part of the ruler, and loyalty on that of the minister".

"Li Yun" (《禮運》) summarizes human ethics in terms of five relationships: father and son, brother, husband and wife, elder and younger and ruler and minister. Similarly, "Mencius" summarizes human relations as "father and son have relatives, monarch and minister have righteousness, husband and wife are different, seniors and children are orderly, and friends have trust." Similarly, in Chapter of "Tengwengong Shang" (《滕文公》) in the book of Mengzi (《孟子》), human ethics is refined as "Father and son have kinship, ruler and minister have righteousness, husband and wife have distinction, seniority and childhood are in order, and friends have trust." The Five Relationships (五倫) in "Li Yun" has "elder and younger" but no "friends," while in "Mencius," it has "friends" but no "brothers," and "brothers" may be included in the "elder and younger" relationship.

The Book of Zhongyong (中庸) summarizes the Five Relationships in terms of father and son, ruler and minister, husband and wife and brother and friend, which has been widely accepted in later times. What is highlighted in the Five Relationships are the reciprocal and differentiated responsibilities of the two parties to the interpersonal relationship, i.e., the Five Relationships place ethical demands on each of the parties to the interpersonal roles. For example, the father's duty to be loving to his children corresponds to the child's duty to be filial to his father.

Although there are more than these five kinds of interpersonal relationships, the Five Relationships constitute the most basic human relationship after all, and other interpersonal relationships can be either reducible to or extend from the Five Relationships. However, all the provisions of the Five Relationships are the entirety of Ren Lun (人倫). Strictly speaking, Ren Lun include two dimensions: one is the Five Relationships (五倫), which define people's different roles, positions, and corresponding responsibilities in society, family, and politics; and the other is The Wu Chang (五常), including the five virtues of benevolence, righteousness, propriety, wisdom, and faith, define the characteristics that make a human being different from other beings. This is the content of the universal human nature understood by Confucianism, which transcends any specific era and specific societies. The Five Relationships is a person's "Wei Fen" (位分) in society, while the Five Chang is his "Xing Fen" (性分) in the cosmos (Y. Chen 2019b, pp. 29–41).

The differentiation between the Five Relationships (五倫) and the Five Chang (五常) in Ren Lun (人倫) is an essential aspect of Ren Lun that differs from from Zong Fa (宗法), which, as a form of organization, is intended to integrate individuals into different levels of patriarchal community. This is essentially a hierarchical community centered on the king, in which the subordination of human beings to the membership of the community is emphasized, rather than the nature of human beings as human beings; although Zong Fa (宗法) can develop interpersonal relationships corresponding to the Five Relationship, it is difficult to develop the Five Chang as universal human beings.

Moreover, there is a qualitative difference between the interpersonal relationship model of the Five Relationships and the Zong Fa system. The former is centered on the small family, while the latter is centered on the clan to establish an individual's identity and position in a political-ethical society; moreover, in the Five Relationships, the ethical dimension takes precedence over the political dimension, while the Zong Fa system has a political function that takes precedence over the ethical function. Among the essential differences between the Zong Fa and the Five Relationships, there is another point that deserves attention: although the Zong Fa is based on the name of fathers, ancestors, etc., it is the political brotherhood that becomes the primary concern, whereas the first of the Five Relationships is the father–son relationship in the family. The transformation from Zong Fa (宗法) to Ren Lun (人倫) has transformed the fundamental meaning of filial piety. It is no longer the reverence of brothers of the same clan for the dead common ancestors but rather the filial piety of children to their parents.

Moreover, filial piety runs through their parents' lives and continues after their deaths. The reason why the relationship between benevolence and filial piety constitutes a fundamental issue in Confucian thought on Ren Lun (人倫) lies in the fact that the two centrally manifest the relationship between the Five Chang (五常) and the Five Relationships (五倫), and when the two are integrated into each other, then the relationship between benevolence and filial piety, between the Five Chang and the Five Relationship, is no longer either one or the other; rather, there is an element of you in me, and me in you, and the requirement to become a certain kind of social role and the requirement to become a human being, as the two dimensions of Ren Lun, are combined together here.

The requirement to become a particular role, such as a son of some kind, combined with the requirement to become a human being, together construct the Confucian way of filial piety. The essence of filial piety is not simply to obey and comply with one's parents' wishes. Absolute obedience to parents is not directly related to filial piety, nor is it the key aspect of being filial. The true essence of filial piety lies not in blindly obeying parents but rather in offering respectful advice and counsel in a gentle and considerate manner when there are differences or disagreements. *Book of Rites*, "Ji Yi" (《禮記●祭義》), emphasizes: "What the superior man calls filial piety requires the anticipation of our parents' wishes, the carrying out of their aims and their instruction in the path (of duty)." This is consistent with the statement of Confucius recorded in the *Analects*, "Li Ren" (《論語●裏仁》): "In serving his parents, a son may remonstrate with them, but gently; when he sees that they

do not incline to follow his advice, he shows an increased degree of reverence, but does not abandon his purpose; and should they punish him, he does not allow himself to murmur".

Indeed, advising and remonstrating with one's parents to prevent them from engaging in unjust actions is an integral part of filial piety. The true essence of filial piety lies in carrying on the aspirations of one's parents (以子繼父). The meaning of "carrying on" (繼) in this context has been pointed out in *Book of Rites*, "Zhong Yong" (《禮記●中庸》), as: "Now filial piety is seen in the skillful carrying on the wishes of our forefathers, and the skillful carrying forward of their undertakings." The essence of filial piety lies in carrying on the aspirations, carrying forward of the undertakings and continuing the legacy of our ancestors, ensuring that their unfulfilled aspirations and accomplishments are not lost, and further developing and glorifying them. In this way, it becomes the inherent meaning of filial piety to bring honor to one's ancestors and family. In other words, what is emphasized here is not only the sense of time and historical consciousness that is generated through generations but also the awareness of historical continuity between successive generations. Through this awareness, individuals are integrated into the familial process of historical continuity, and the unity of "father and son" is realized through subjective aspirations and objective endeavors.

Originally, filial piety referred to the relationship between children and their parents and ancestors. However, through this vertical relationship, the subject of filial piety has shifted toward the individual's relationship with oneself. *The Classic of Filial Piety* (《孝經》) states: "Our bodies—to every hair and bit of skin—are received by us from our parents, and we must not presume to injure or wound them. This is the beginning of filial piety. When we have established our character by the practice of the (filial) course, so as to make our name famous in future ages and thereby glorify our parents, this is the end of filial piety." The starting point of filial piety lies in respecting one's own body and life, while the ultimate goal of filial piety is to establish oneself and leave a positive legacy for future generations. Whether at its starting point or its endpoint, filial piety becomes a requirement that children impose on themselves. This requirement does not imply absolute obedience to parents, but rather directs individuals toward self-realization in terms of personal character and conduct. In other words, although filial piety appears to be rooted in familial relationships within the household, it is elevated to a universal spiritual dimension that defines the essence of being human.

Whereas emphasis on filial piety can be observed within the framework of the order of Zong Fa (宗法), it is important to note that in this context, filial piety is not directed toward children's reverence for their parents but rather toward the religious-like awe that the living have for their deceased ancestors. In the context of the Western Zhou Dynasty, filial piety was not primarily an ethical principle for maintaining the father–son relationship, but rather a responsibility of the rulers and the eldest sons of the main lineage, with the purpose of "honoring ancestors, respecting the ancestral temple" and "preserving the clan and maintaining the family." In the communal structure of the Western Zhou Dynasty, the term "filial son" referred to the legitimate sons of the lineage rather than the illegitimate ones. During the Western Zhou period, filial piety was directed toward ancestral spirits and not toward living individuals. When discussing filial piety toward fathers, it was understood as an extension of the broader principle of honoring ancestors and respecting the ancestral temple: " To fulfill filial piety towards ancestors" ("Chou Er Zhong" 《儔兒鐘》), "To fulfill filial piety towards deceased parents or ancestors" (Xi Zhong Zhong 《兮仲鐘》), "To fulfill filial piety towards previous generations who possessed cultural and moral virtues" (*Book of History*, "Wen Hou Zhi Ming" 《尚書·文侯之命》), "It was to show the filial duty which had come down to him" (*Book of Songs*, "Wen Wang You Sheng" 《詩經·文王有聲》).

In the Zong Fa (宗法) community, the fundamental principle of filial piety was primarily focused on showing respect to ancestors (尊尊) rather than immediate family members (親親). The concept of "filial piety not towards one's father" ("無父之孝"), as discussed by Mozi, was its essence that remained (Zha 2006, pp. 10–97). The following statement

from *The Classic of Filial Piety* (《孝經》) also bears the imprint of the concept of filial piety within the order of Zong Fa (宗法): "…To preserve the altars of their land and grain is the filial piety of the princes of states. …To preserve their ancestral temples is the filial piety of high ministers and great officers. …To preserve their emoluments and positions, and to maintain their sacrifices is the filial piety of inferior officers." It is evident that the filial piety emphasized within the order of Zong Fa (宗法), with its focus on respecting ancestors and fulfilling social responsibilities, differs significantly from the filial piety advocated by Confucius and Mencius, which emphasizes the importance of parent–child relationships. The change of the object of filial piety from the dead ancestors in the Zong Fa society to the living father in the family corresponds to the structural changes in society since the Spring and Autumn Period and the Warring States Period. Luo Tai's research on tombs shows that with the decline of the primacy of Zong Fa organizations, the importance of ancestor worship decreased during the Spring and Autumn and Warring States Periods, the tree-like structure of the patriarchal society gradually transformed into a loosely knit society and the rites, which were used in the Western Zhou to maintain the patriarchal organization, were transformed into an ecumenic ethical regime. In this context, filial piety is no longer a way of political cohesion and community building but an expression of personal virtue (von Falkenhausen 2006, pp. 290–301, 397–99).

### 4. From the Principle of Qin Qin Zun Zun (Emphasis on Immediate Family Relationships and Respect for Ancestors親親尊尊) to Ren Yi (Benevolence and Righteousness仁義): The Continuity and Transformation of the Order of the Community

In response to the prominent role of brotherhood within the order of the Zong Fa (宗法) community, the basic principles of the order were "Qin Qin" (親親emphasizing the importance of immediate family relationships) and "Zun Zun" (尊尊emphasizing the reverence for ancestors). This is also the spiritual foundation of the Zhou ritual system (周禮). In "Discussions on Yin and Zhou Institutions," Wang Guowei argued that the rituals and institutions of the Western Zhou Dynasty were derived from "Qin Qin" (親親) and "Zun Zun" (尊尊) (G. Wang 2009, p. 314).

Under the political structure of a family-centric system ("家天下"), the principle of Qin Qin (親親) ensured the familial nature of the ruling group. This was achieved through practices such as the enfeoffment of noble descendants and granting surnames and clan names. These measures aimed to unite the Zhou clans within quasi-blood-related families. For ruling members who did not have direct family ties, family connections were established through means such as intermarriage.

In this sense, the principle of "Qin Qin" (親親) facilitated political and social cohesion and unity through the connecting function of family ties. *Yi Li*, "Jin Li" states: "In a large state with the same surname, one would be called Bo Fu (伯父) to those of the same surname; for those of a different surname, one would be called Bo Jiu (伯舅). In a small state with the same surname, one would be called Shu Fu (叔父); for those of a different surname in a small state, one would be called Shu Jiu (叔舅)."[6] It can be seen from this that the principle of "Qin Qin" (親親) establishes a strong connection between Tian Zi (天子) and the feudal lords. The foundational significance of the principle of "Qin Qin" (親親) in the system of Zong Fa (宗法) can also be observed through the mourning rituals (system of Sang Fu喪服制度) that are closely related to the order of Zong Fa (宗法). Wu Chengshi regarded the statement "in the case of the nearest kindred, there is a break in it at the end of a year" ("Zhi Qin Yi Qi Duan" 至親以期斷) in the *Book of Rites*, "San Nian Wen" (《禮記●三年問》) as the "the principle of principles" in mourning rituals (system of Sang Fu喪服制度). The phrase "Zhi Qin Yi Qi Duan" (至親以期斷) refers to the duration of mourning, which is based on a period of one year as the basic unit. The five stages of mourning are centered around close relatives, with the mourning period as the main axis, and additional rituals or changes in mourning practices are determined accordingly (Wu 1998, pp. 316–29). This mourning ritual (system of Sang Fu喪服制度) itself demonstrates the importance of the principle of "Qin Qin" (親親) in the order of Zong Fa (宗法). In a certain sense, the order

of Zong Fa (宗法) can be seen as a kinship community formed by the aggregation of the principle of Qin Qin (親親).

However, within the kinship community constructed by the principle of Qin Qin (親親), the principle of Zun Zun (尊尊) takes precedence as the first-order principle. As a result, situations often arise where the respect shown to a clan elder surpasses the affection between father and son. The principle of Qin Qin (親親) between father and son gives way to the hierarchical principle of Zun Zun (尊尊) among clan members. In terms of kinship relations, both the legitimate sons (宗子) and the illegitimate sons (庶子) are considered close relatives, sharing the same father. They should be equal as father and son, but their respective legitimacy status determines the hierarchy among brothers.

This can be observed from the mourning rituals closely associated with the order of Zong Fa (宗法). The father mourns for the eldest son for three years, while mourning for the other children lasts for one year. This illustrates that the legitimacy status creates distinctions of hierarchy among brothers. For someone who belongs to a large clan without an heir and adopts a son to carry on the lineage, their biological father may indeed be a close relative, but compared to the adoptive father who ensures the continuation of the lineage, the biological father's mourning period is downgraded, while the adoptive father receives the most solemn mourning attire. The biological father is thus considered inferior to the adoptive father. Within the framework of a family-centric system ("家天下"), it is necessary to strengthen the principle of Zun Zun (尊尊) to establish order within the Zong Fa (宗法) community.

The principle of Qin Qin (親親) and Zun Zun (尊尊) in the order of Zong Fa (宗法) shares a historical continuity with the early Confucian emphasis on Ren Yi (benevolence and righteousness仁義) as the core principles of humanity. *Book of Rites*, "Da Zhuan" (《禮記●大傳》), states: "Where the starting-point was in affection, it began with the father, and ascended by steps to the ancestor. Where it was in a consideration of what was right, it began with the ancestor, and descended in natural order to the deceased father. Thus the course of humanity (in this matter of mourning) was all comprehended in Qin Qin (親親)."

This is a direct correlation between Ren (benevolence仁) and Qin Qin (親親), and between Yi (righteousness義) and Zun Zun (尊尊). The *Book of Rites*, "Zhong Yong" (《禮記●中庸》), expresses it more directly: "Ren (benevolence仁) is the characteristic element of humanity, and the great exercise of it is in loving relatives (親親). Yi (righteousness義) is the accordance of actions with what is right, and the great exercise of it is in honoring the worthy. The decreasing measures of the love due to relatives, and the steps in the honor due to the worthy, are produced by the principle of propriety".

Based on the above discussion from the *Book of Rites*, Guo Songtao pointed out: "People in both large and small lineages hold their ancestors in high esteem as the source of their existence. Yi (righteousness義) is closely connected to ancestors, which is why ancestors are considered significant. Ren (benevolence仁) is discussed in terms of mourning rituals, while Yi (righteousness義) is discussed in terms of the ancestral system" (Guo 1992, pp. 430–31). The interplay between the Zong Fa (宗法) system and ritual practices signifies the intertwined nature of the principles of Qin Qin (親親) and Zun Zun (尊尊). Wang Guowei emphasizes: "The people of the Zhou Dynasty employed the principle of Zun Zun (尊尊) to manage the principle of Qin Qin (親親). This was reflected in the establishment of the system of distinguishing between the legitimate and illegitimate, where the status of legitimate and illegitimate sons was determined based on their relationship to the father. Conversely, they also managed the principle of Zun Zun (尊尊) through the principle of Qin Qin (親親), which was reflected in the establishment of the ancestral temple system. These interrelated principles formed the basis for the Zhou people's establishment of the system of rituals and music" (G. Wang 2009, p. 313).

However, it does not mean that the principles of Zong Fa (宗法) and the mourning ritual are based on the principles of Ren Yi (仁義). It is simply stating that the early Confucian principles of Ren Yi (仁義) are actually a further refinement and purification of the

principles of Qin Qin (親親) and Zun Zun (尊尊) within the system of ancestral worship. The order of Qin Qin (親親) and Zun Zun (尊尊) are principles that are embodied through the system, and they are always intertwined with the Zhou Dynasty's ritual system and cannot be artificially separated. On the other hand, Ren Yi (仁義) are spiritual principles that go beyond specific institutions. Correspondingly, adhering to the principles of Qin Qin (親親) and Zun Zun (尊尊) is actually following the Zhou rituals that have permeated both the customs and institutional levels. On the other hand, starting from the principles of Ren Yi (仁義) is based on values and moral principles, extending to the secular and non-institutional level of individual autonomy in later generations. Therefore, Qin Qin (親親) and Zun Zun (尊尊) are ways of incorporating individuals into the ancestral system and the entire Zhou ritual. In contrast, the principle of Ren Yi (仁義) transcends the dimensions of customs and institutions.

Whereas the principles of Ren Yi (仁義) originated from the ancestral system of the Zhou Dynasty, once established, they possess a universal significance that goes beyond the confines of the order of Zong Fa (宗法). In fact, the unfolding of this universalization occurs through the liberation from the binding of Qin Qin (親親) and Zun Zun (尊尊) associated with the current order of Zong Fa (宗法). Ren (仁) is the treatment of individuals as human beings, and it shares a continuity with the principle of Qin Qin (親親).

This has been recognized by various schools of Confucian thought throughout history: Ling Tingkan 凌廷堪 has profoundly revealed the relationship between the way of benevolence and righteousness and the system of mourning clothes (Ling 2009, pp. 15–16). Zhang Shouan 张寿安 also has a lot of inventions about this. She argues that the foundation of the Confucian ritual order is the tension structure between Qin Qin (亲亲) and Zun Zun (尊尊), and that both thought and system depend on the balance of Qin Qin and Zun Zun to be stable; more importantly, Qin Qin and Zun Zun are the concrete implementation of benevolence and righteousness (Zhang 2005, p. 138). Zeng Guofan 曾国藩 emphasized more clearly: "The former kings created the ritual system, by beautifying and tempering people's love, to make rituals in harmony with benevolence, and to establish hierarchical institutions to express people's awe, so that rituals are in harmony with righteousness, although the content of rituals it is ever-changing, but fundamentally it is based on benevolence and righteousness" (Zeng 2011a, pp. 216–18). In Zeng Guofan's view, not everyone can understand benevolence and righteousness, and this was fully taken into account when the ancient kings established rituals, so benevolence and righteousness were internalized into the customs, rituals and systems of the living world, so that people could be infected by institutionalized benevolence and righteousness without realizing it (Zeng 2011b, p. 175). In this sense, benevolence and righteousness are not only the foundation of the ritual system but also the end of the ritual system.

However, it is equally important to recognize their differences. The principle of Qin Qin (親親) applies only to blood relatives up to the sixth generation, while Ren (仁) implies a love that extends to every individual or all humanity. Although there may be hierarchical differences in the way love is expressed, treating individuals as human beings is inherently impartial. In a broader sense, Ren (仁) can encompass Qin Qin (親親), as Qin Qin (親親) is just one aspect of loving others. The concept of "Qin Qin, Ren Min" (be affectionate to his parents, and lovingly disposed to people generally 親親、仁民) mentioned by Mencius can both be included within the concept of Ren (仁). In the same way, Zun Zun (尊尊) is out of respect for the revered, and the revered is considered honorable.

However, Yi (義), as a principle of what is morally right and appropriate, can encompass the principle of Zun Zun (尊尊) within itself. This means that the principle of Zun Zun (尊尊) can be subsumed under the principle of Yi (義). Consequently, Ren Yi (仁義), as humane principles within the order of the Ren Lun (人倫) community, signify the fundamental principles that define humanity. The specific manifestations of Qin Qin (親親) and Zun Zun (尊尊) within the order of Zong Fa (宗法) become expressions of these universal principles within a particular social structure, such as the Zhou Dynasty's Zong Fa (宗法) society.

     In this way, the principle of Ren Yi (仁義) transcends the principle of Qin Qin (親親) and Zun Zun (尊尊), becoming an independent principle beyond the existing political and social system. This demonstrates both the continuity and the differences between the social order of Ren Lun (人倫) and the order of Zong Fa (宗法). In short, Qin Qin (親親) and Zun Zun (尊尊), as the foundations of the order of Zong Fa (宗法), still serve as ways to integrate individuals into the community, while Ren Yi (仁義) point toward universal human nature. Here, one can find a universal human consciousness that goes beyond specific communities and societies. This implies that within the order of the Zongfa (宗法) community, becoming a person is synonymous with becoming a member of the community, without any differentiation. However, within the order of order of the Ren Lun (人倫) community, there is a differentiation between becoming a person and becoming a member of the community. If Xiao (filial piety孝) is seen as the virtue that integrates the individual into the community, then Ren (benevolence仁) represents the universal essence of being human. It is no longer tied to the status of being a community member.

     In conclusion, the transition from the order of the Zong Fa (宗法) community of the Western Zhou Dynasty to the order of the Ren Lun (人倫) community in the early Confucianism of Confucius and Mencius signifies a paradigm shift in the order of society. This shift is closely related to structural changes within the political and social sphere, as the basic unit of society shifted from the clan to the nuclear family. Consequently, the contemplation and construction of order had to seek new starting points. Early Confucianism laid the intellectual foundation for the social order of the Ren Lun (人倫) community. Over the following two thousand years, although the family remained the foundation of the social structure and the traditional Chinese political system operated under the framework of a family-centric system ("家天下"), these two elements provided institutional support for the social order of the Ren Lun (人倫) community.

     In turn, objectively speaking, early Confucianism played a role in laying the foundation for the traditional order over the course of two millennia. Simultaneously, the moral virtue of the Ren Lun (人倫) community, which is Ren (仁), and the ethical virtue of filial piety are combined within the structure of the community. This intertwining of the construction of community order and universal human nature occurs on the premise of differentiation.

**Funding:** This research received no external funding.

**Institutional Review Board Statement:** Not applicable.

**Informed Consent Statement:** Not applicable.

**Data Availability Statement:** This study did not create new data, but I'm happy to share this article at MDPI journals.

**Conflicts of Interest:** The author declares no conflict of interest.

## Notes

[1]   Please refer to Xue Xuan (薛瑄)'s summary of the functions of Zong Fa (宗法). See (Li 2010, p. 311).

[2]   *Book of Rites*, "Sang Fu Xiao Ji" (《禮記●喪服小記》). According to the interpretation of Zheng Xuan(鄭玄), "One's own father is positioned above oneself in the hierarchy, while one's children are positioned below. Together, they form three generations. Expanding from the father to the grandfather and from the children to the grandchildren constitutes five generations. Expanding from the grandfather to the great-grandfather and from the grandchildren to the great-grandchildren and beyond, it amounts to a total of nine generations". The word "殺" means "as the blood relation becomes more distant, the level of mourning attire gradually decreases".

[3]   Please refer to "Great Commentary (《大傳》)" of *The Book of Rites* (《禮記章句》) in his book.

[4]   In Chapter "Wei Zheng" (《為政篇》) of *The Analects of Confucius* (《論語》), this statement is quoted with the word "Hu" ("乎"), but the version annotated by Huang Kan (皇侃本) and the version carved on stone during the Han Dynasty (漢石經本) quotes this statement with the word "Yu" ("於").

[5]   Patriarchal-based Zong Fa (宗法) was one form of ethical life in the Western Zhou period, but in turn, ethics was not equal to or limited to patriarchal-based patriarchy, but had broader connotations.

6    Similar to this, in *Book of Rites*, "Qu Li◉" (《禮記●曲禮下》) it is stated: "Chief among the five officers are Bo (伯), to whom belong the oversight of quarters. If they are of the same surname as Tian Zi (天子), he styles them 'Bo Fu (伯父)', if of a different surname, 'Bo Jiu (伯舅)'…The head prince in each of the nine provinces… If he be of the same surname as himself, Tian Zi (天子) calls him 'Shu Fu (叔父)', if he be of a different surname, 'Shu Jiu (叔舅)'".

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
