# Peer review of "From the Order of Zong Fa (宗法) to the Order of Ren Lun (人倫)—Confucianism and the Transformation of the Paradigm of Early Chinese Communities"

_religions, doi:10.3390/rel14091091_

Round 1
Reviewer 1 Report
This is one of the few well-written articles I have read on the study of early Chinese community theory. In the article, the author takes the rise of Confucianism as a boundary and examines very thoroughly how community theory in the Zhou dynasty was transformed from a “patriarchal” (zongfa 宗法) model to an “ethical” (renlu 人倫) one. The detailed and cogent arguments in the article imply that the author has better academic credentials in the aforementioned research field.
Throughout the article, it can be seen that the author makes a clear distinction between the Confucian “ethical” model and the “patriarchal” model constructed at the beginning of the Zhou dynasty. However, my concern is that, in a broad sense, the former (“ethical”) has its roots in the latter (“patriarchal”), and therefore there should be an overlap between the two in in terms of connotation, so I suggest that the authors add a note at the appropriate place in the article to briefly clarify this point.
Some minor spell checking may be required, such as the repeated use of the definite article “the” before the noun “order” in line 25, one of which should be deleted.
Author Response
Please see footnote 11. (A note has been added that reads as follows: Patriarchal-based Zong Fa(宗法) was one form of ethical life in the Western Zhou period, but in turn, ethics was not equal to or limited to patriarchal-based patriarchy, but had broader connotations.)
Some minor spell checking may be required, such as the repeated use of the definite article “the” before the noun “order” in line 25, one of which should be deleted. (deleted)
Reviewer 2 Report
The manuscript presents an interesting exploration of filial piety and fraternal love within the context of the Zhou Dynasty, and how these concepts were integrated into the social system of rites and music. The author also discusses the concept of Qin Qin and Ren, and how these concepts apply to familial and broader human relationships. However, I have identified several areas where the manuscript could be improved to enhance its clarity, coherence, and scholarly rigor.
**Major Comments:**
1. **Clarity and Precision of Language**: The manuscript could benefit from more precise language and clearer explanations. For example, on page 5, lines 273-282, the author discusses the importance of familial relationships and their impact on societal structures. However, the language used is somewhat convoluted and could be simplified for better understanding. I suggest revising the sentence structure and using more straightforward language to convey the same ideas.
2. **Cohesion and Flow**: The manuscript seems to jump between different ideas without clear transitions. For example, on page 6, lines 324-335, the author discusses the decline of the major lineage system and the rise of the nuclear family. However, the transition between these two ideas is not smooth. I recommend using transitional phrases or sentences to guide the reader from one idea to the next.
3. **Contextualization**: The manuscript could do a better job of situating its arguments within the broader scholarly conversation. For example, on page 11, lines 574-583, the author discusses the concepts of Qin Qin and Ren, but it's not clear how these concepts relate to the existing literature on the topic. I suggest providing more context for the arguments being made.
4. **Conclusion**: The conclusion of the manuscript (page 7, lines 376-389) could be strengthened by summarizing the main arguments more clearly and discussing the implications of the research. I recommend revising the conclusion to clearly restate the main arguments and discuss their significance.
**Minor Comments:**
1. **Interpretation of Historical Texts**: The author relies heavily on historical texts and inscriptions to support their arguments. However, these sources are open to interpretation, and the author's interpretation may not be the only or most accurate one. I suggest the author provide more evidence to support their interpretation of these historical texts and inscriptions, or discuss alternative interpretations and explain why they believe their interpretation is the most accurate.
2. **Conceptual Definitions**: The author's definitions of Qin Qin and Ren may be open to debate. The author could provide more evidence to support their definitions of Qin Qin and Ren, or discuss alternative definitions and explain why they believe their definitions are the most accurate.
3. **Causal Relationships**: The author suggests causal relationships between various concepts. However, these causal relationships may not be as clear-cut as the author suggests, and other factors may also play a role. I recommend the author provide more evidence to support their proposed causal relationships, or discuss other potential factors that could influence these relationships.
In conclusion, while the manuscript presents an interesting exploration of filial piety and fraternal love in the Zhou Dynasty, it requires significant revisions to enhance its clarity, coherence, and scholarly rigor. I believe that addressing these issues will greatly improve the manuscript and make it a valuable contribution to the field.
Author Response
**Clarity and Precision of Language**: The manuscript could benefit from more precise language and clearer explanations. For example, on page 5, lines 273-282, the author discusses the importance of familial relationships and their impact on societal structures. However, the language used is somewhat convoluted and could be simplified for better understanding. I suggest revising the sentence structure and using more straightforward language to convey the same ideas.(Revised. Please see lines 282-292 in the PDF version, i.e. lines 273-282 in the original version. )
**Cohesion and Flow**: The manuscript seems to jump between different ideas without clear transitions. For example, on page 6, lines 324-335, the author discusses the decline of the major lineage system and the rise of the nuclear family. However, the transition between these two ideas is not smooth. I recommend using transitional phrases or sentences to guide the reader from one idea to the next.
【Response】Please see line341-371. The following paragraph has been added:
The background of the establishment of the Ren Lun(人倫) community is the disintegration of the Zong Fa community. As the creation of the Western Zhou Dynasty, the Zong Fa was closely associated with the feudal system. The feudal system of the Zhou was divided into two levels: first, the king of the Zhou divided the children of the royal family into vassal states, the son of heaven gave them a surname姓, and the recipients of the title became vassals by acquiring a surname and establishing a state; and second, the vassals were given shi(氏)(a branch of the family name), and the recipients were given land to establish a Aristocrat's House and became ministers. The condition for the possibility of feudalism was that the Zhou people continued to expand, so there was surplus land and population, which could be distributed to the sons and younger brothers of the king; on the other hand, the population to be rewarded was based on the clan. The grantee received a rewarding surname or shi(氏) from a superior ruler and thus became a legitimate eldest son(宗子)in a patriarchal Zong Fa(宗法) community. In other words, the eldest son inheritance system in the feudal system is actually the application of the patriarchal Zong Fa system, so the patriarchal system is also called the Zong Fa feudal system.Once the Zhou had no surplus land and population to distribute, and once the basic unit of society had diverged from the patriarchal clan to the small main family, then the patriarchal community and its ethical form of life became untenable. In fact, during the Spring and Autumn and Warring States periods in which Confucius and Mencius lived, the patriarchal clan system had declined, and the backbone family, with five or seven members as the main body, constituted the basic unit of society. The monogamous family, formed by ordinary men and women, came to the forefront of history, and with the establishment of the system of the "bianhuqimin"編戶齊民, this kind of small family constitutes the foundation of society. While the Zong Fa system demonstrated the political connection between the old brother's state and the young brother's state, the human relationships within the main family lost their political function of unifying the different states and became an ethical field for social ordering. For example, filial piety is no longer a way for people in different families within the same clan in the patriarchal system to remember their common ancestor, but a way for small families to respect for patrilineal parents of the family.
**Contextualization**: The manuscript could do a better job of situating its arguments within the broader scholarly conversation. For example, on page 11, lines 574-583, the author discusses the concepts of Qin Qin and Ren, but it's not clear how these concepts relate to the existing literature on the topic. I suggest providing more context for the arguments being made.
【Response1】
After this paragraph (“This has been recognized by various schools of Confucian thought throughout history”,p12,line574), the following paragraph has been added, please see lines 696-713.
Ling Tingkan凌廷堪has profoundly revealed the relationship between the way of benevolence and righteousness and the system of mourning clothes. Zhang Shouan张寿安 also has a lot of inventions about this. She argues that the foundation of the Confucian ritual order is the tension structure between Qin Qin(亲亲) and Zun Zun(尊尊), and that both thought and system depend on the balance of Qin Qin and Zun Zun to be stable;more importantly, Qin Qin and Zun Zun are the concrete implementation of benevolence and righteousness. Zeng Guofan曾国藩 emphasized more clearly: "The former kings created the ritual system, by beautifying and tempering people's love, to make rituals in harmony with benevolence, and to establish hierarchical institutions to express people's awe, so that rituals are in harmony with righteousness, although the content of rituals it is ever-changing, but fundamentally it is based on benevolence and righteousness.” In Zeng Guofan’s view, not everyone can understand benevolence and righteousness, and this was fully taken into account when the ancient kings established rituals, so benevolence and righteousness were internalized into the customs, rituals and systems of the living world, so that people could be infected by institutionalised benevolence and righteousness without realising it. In this sense, benevolence and righteousness are not only the foundation of the ritual system, but also the end of the ritual system.
Additional literature supplements, including archaeological studies and literature based on archaeological research, please see lines 575-584:
“The change of the object of filial piety from the dead ancestors in the Zong Fa society to the living father in the family corresponds to the structural changes in society since the Spring and Autumn Period and the Warring States Period.Luo Tai’s research on tombs shows that with the decline of the primacy of Zong Fa organizations, the importance of ancestor worship decreased during the Spring and Autumn and Warring States Periods, the tree-like structure of the patriarchal society gradually transformed into a loosely knit society, and the rites, which were used in the Western Zhou to maintain the patriarchal organization, were transformed into an ecumenic ethical regime. In this context, filial piety is no longer a way of political cohesion and community building, but an expression of personal virtue.”
**Conclusion**: The conclusion of the manuscript (page 7, lines 376-389) could be strengthened by summarizing the main arguments more clearly and discussing the implications of the research. I recommend revising the conclusion to clearly restate the main arguments and discuss their significance.
【Response1】The following portion has been added, please see lines 418-480.
"Li Yun"(《禮運》) summarizes human ethics in terms of five relationships five relationships: father and son, brother, husband and wife, elder and younger, and ruler and minister. Similarly, "Mencius" summarizes human relations as "father and son have relatives, monarch and minister have righteousness, husband and wife are different, seniors and children are orderly, and friends have trust".Similarly, in Chapter of “Tengwengong Shang" (《滕文公》)in the book of Mengzi(《孟子》), human ethics is refined as "Father and son have kinship, ruler and minister have righteousness, husband and wife have distinction, seniority and childhood are in order, and friends have trust".The Five Relationships(五倫) in "Li Yun" has " elder and younger " but no "friends", while in "Mencius" it has "friends" but no "brothers", and maybe “brothers” are included in the “elder and younger” one. The Book of Zhongyong(中庸) summarises the Five Relationships in terms of father and son, ruler and minister, husband and wife, brother and friend, which has been widely accepted in later times. What is highlighted in the Five Relationship are the reciprocal and differentiated responsibilities of the two parties to the interpersonal relationship, i.e., the Five Relationship place ethical demands on each of the parties to the interpersonal roles. For example, the father's duty to be loving to his children corresponds to the child's duty to be filial to his father.Although there are more than these five kinds of interpersonal relationships, the Five Relationship constitute the most basic human relationship after all, and other interpersonal relationships can be either reducible to or extend from the Five Relationship.However, all the provisions of the Five Relationship are the entirety of Ren Lun(人倫). Strictly speaking, Ren Lun include two dimensions: one is the Five Relationship(五倫), which define people's different roles, positions, and corresponding responsibilities in society, family, and politics; and the other is The Wu Chang(五常), including the five virtues of benevolence, righteousness, propriety, wisdom, and faith, define the characteristics that make a human being different from other beings. This is the content of the universal human nature understood by Confucianism, which transcends any specific era and specific societies. While the Five Relationship is a person's "Wei Fen"(位分) in society, while the Five Chang is his "Xing Fen"(性分) in the cosmos. The differentiation between the Five Relationship(五倫) and the Five Chang(五常) in Ren Lun(人倫) is an essential aspect of Ren Lun that differs from from Zong Fa(宗法) which, as a form of organization, is intended to integrate individuals into different levels of patriarchal community, which is essentially a hierarchical community centred on the king, and in which the subordination of human beings to the membership of the community is emphasised, rather than the nature of human beings as human beings, so that even though Zong fa(宗法) can develop interpersonal relationships corresponding to the Five Relationship, it is difficult to develop the Five Chang as universal human beings. What's more, there is a qualitative difference between the interpersonal relationship model of the Five Relationship and the Zong Fa system. The former is centered on the small family, while the latter is centered on the clan to establish an individual's identity and position in a political-ethical society; moreover, in the Five Relationship, the ethical dimension takes precedence over the political dimension, while the Zong Fa system has a political function that takes precedence over the ethical function.Among the essential differences between the Zong Fa and the Five Relationships, there is another point that deserves attention: although the Zong Fa is based on the name of fathers, ancestors, etc., it is the political brotherhood that becomes the primary concern; whereas the first of the Five Relationships is the father-son relationship in the family.The transformation from Zong Fa(宗法) to Ren Lun(人倫) has transformed the fundamental meaning of filial piety. It is no longer the reverence of brothers of the same clan for the dead common ancestors, but rather the filial piety of children to their parents. Moreover, filial piety runs through their parents’ lives and after their deaths.The reason why the relationship between benevolence and filial piety constitutes a fundamental issue in Confucian thought on Ren Lun(人倫) lies in the fact that the two centrally manifest the relationship between the Five Chang(五常) and the Five Relationship(五倫), and when the two are integrated into each other, then the relationship between benevolence and filial piety, between the Five Chang and the Five Relationship, is no longer either one or the other, but rather, there is an element of you in me, and me in you, and the requirement to become a certain kind of social role and the requirement to become a human being, as the two dimensions of Ren Lun, are combined together here.
【Response2】The following has been deleted, please see lines481-500:Indeed, it is evident that no one can exempt themselves from the requirements of the order of human relationships, regardless of the role they occupy. However, in the father-son relationship, filial piety tends to be emphasized. Some of the Confucian disciples advo-cate that “Filial piety and fraternal submission. Are they not the root of all benevolent actions?”(The Analects, “Xue Er”《論語·學而》)Here, filial piety is regarded as the foundation of benevolent actions, and “benevolent actions” implies that individuals col-lectively become humane. Filial piety is thus elevated as the fundamental path to be-coming a mature person. This understanding reinforces the foundational position of the family in the order, as the family’s essence lies within the order of Ren Lun(人倫). The order of Ren Lun(人倫), in turn, centers around the father-son relationship, and the key to maintaining this relationship lies in the son’s filial piety. Filial piety from children to parents is not only a way to repay the gratitude for their upbringing but also a form of reciprocal gesture based on the bond of blood relation. More importantly, in early Con-fucianism, it was elevated to a principle of humaneness. Due to the special significance of the family in one’s growth, filial piety is regarded as the foundation of all moral educa-tion and ethics. Moreover, The Classic of Filial Piety(《孝經》) further elevates filial piety to unprecedented heights: “Now filial piety is the root of (all) virtue, and (the stem) out of which grows (all moral) teaching," “filial piety is the constant (method) of Heaven, the righteousness of Earth, and the practical duty of Man”.
【Response3】Also, “The requirement to become a particular role such as a son of some kind, combined with the requirement to become a human being, together construct the Confucian way of filial piety” has been added, please see lines 501-503.
**Minor Comments:**
**Interpretation of Historical Texts**: The author relies heavily on historical texts and inscriptions to support their arguments. However, these sources are open to interpretation, and the author's interpretation may not be the only or most accurate one. I suggest the author provide more evidence to support their interpretation of these historical texts and inscriptions, or discuss alternative interpretations and explain why they believe their interpretation is the most accurate.
【Response】I have made further supplements, elaborations and demonstrations in some places of the article.
**Conceptual Definitions**: The author's definitions of Qin Qin and Ren may be open to debate. The author could provide more evidence to support their definitions of Qin Qin and Ren, or discuss alternative definitions and explain why they believe their definitions are the most accurate.
【Response】The paper is less likely to define Qin Qin(親親)and Ren(仁), but rather to categorise them. As for the definition of these two concepts, there will be no consensus so far and it is not the point of this paper. My aim and concern in this paper is to differentiate between Qin Qin(親親) as a Zong Fa(宗法) principle and Ren(仁) as a Ren Lun(人倫) principle, to reveal their differences and connections, and thus to provide a different aspect from what has already been researched, which is not a a denial of the existing research, but rather an attempt to add new possible approaches to understanding existing problems.
**Causal Relationships**: The author suggests causal relationships between various concepts. However, these causal relationships may not be as clear-cut as the author suggests, and other factors may also play a role. I recommend the author provide more evidence to support their proposed causal relationships, or discuss other potential factors that could influence these relationships.
【Response】 During the specific revision process, I made appropriate supplements in an attempt to clarify the relevant issues more clearly. In addition, some annotations are added. The existing studies given in the annotations have a good discussion on this, and due to space limitations, it is not necessary to list them all.For example, with regard to my point that the Zong Fa pole relationship is dominated by brotherhood, I give specific research literature through notes in my revision.
Reviewer 3 Report
You are obviously deeply conversant with Confucian values, thought, communities and relations with family life in ancient China. You clearly know the primary sources very well indeed, as well as the main Chinese-language research on this topic.
Although your English expression is generally very good indeed, I found it a bit difficult to follow in many places. For instance, the paragraphs are too long and give an often unnecessary impression of denseness and complexity.
I think your central argument could be spelt out a bit more clearly. Your emphasis on the two forms of community, which seem to me an important idea I have not seen in the literature before, is clearly stated. But the details within the argument are quite dense and difficult to follow.
Also, I think the secondary books you cite, which appear to be in Chinese, should have both pinyin and Chinese characters as well as the title translated into English. You put characters in the text when citing particular places in the primary sources, as is appropriate. I think you should also give the Chinese characters for secondary sources cited in the footnotes.
Overall, you have definitely contributed to knowledge in the important area of Confucian values in the communities and family life in ancient China. So I recommend your article for publication, after some minor amendments as noted above.
The English expression is very good but often very dense and difficult to follow. One of the reasons is that the paragraphs are too long and not crisp enough.
Author Response
Although your English expression is generally very good indeed, I found it a bit difficult to follow in many places. For instance, the paragraphs are too long and give an often unnecessary impression of denseness and complexity.
【Response】reparagraphed
I think your central argument could be spelt out a bit more clearly. Your emphasis on the two forms of community, which seem to me an important idea I have not seen in the literature before, is clearly stated. But the details within the argument are quite dense and difficult to follow.
【Response】I have made further supplements, elaborations and demonstrations in some places of the article.
Also, I think the secondary books you cite, which appear to be in Chinese, should have both pinyin and Chinese characters as well as the title translated into English. You put characters in the text when citing particular places in the primary sources, as is appropriate. I think you should also give the Chinese characters for secondary sources cited in the footnotes.
【Response】please see the revised footnotes and references.
Overall, you have definitely contributed to knowledge in the important area of Confucian values in the communities and family life in ancient China. So I recommend your article for publication, after some minor amendments as noted above.
Round 2
Reviewer 2 Report
I am impressed with the authors' revisions and the careful attention they've given to the feedback on their manuscript. Each concern from my initial review has been adeptly addressed, rightly enhancing the quality of the paper. Therefore, its publication in Religions seems appropriate.